# Centromere Protein F Is a Potential Prognostic Biomarker and Target for Cutaneous Melanoma

**DOI:** 10.3390/biomedicines13040792

**Published:** 2025-03-25

**Authors:** Lilu Xie, Kangjie Shen, Chenlu Wei, Jiangying Xuan, Jiayi Huang, Zixu Gao, Ming Ren, Lu Wang, Yu Zhu, Shaoluan Zheng, Chuanyuan Wei, Jianying Gu

**Affiliations:** 1Department of Plastic Surgery, Zhongshan Hospital, Fudan University, Shanghai 200433, China; 19301050317@fudan.edu.cn (L.X.); 20301050281@fudan.edu.cn (J.H.);; 2Department of Plastic and Reconstructive Surgery, Zhongshan Hospital (Xiamen), Fudan University, Xiamen 361015, China; 3Xiamen Clinical Research Center for Cancer Therapy, Xiamen 361015, China

**Keywords:** melanoma, CENPF, E2F3, cell cycle, p53

## Abstract

**Background/Objectives:** Cutaneous melanoma (CM) is a highly aggressive malignancy with poor prognosis, necessitating novel biomarkers and therapeutic targets. Centromere protein F (CENPF), a mitotic regulator, has been implicated in tumor progression, but its role in melanoma remains unclear. This study aimed to investigate the clinical significance, biological function, and regulatory mechanisms of CENPF in melanoma. **Methods:** Public melanoma datasets (GSE46517, GSE3189, and GSE7553) were re-analyzed to identify differentially expressed genes (DEGs). CENPF expression was validated in clinical samples (n = 128), melanoma cell lines, and xenograft models. Functional assays (EdU, CCK-8, colony formation, wound healing, transwell, and flow cytometry) and bioinformatics analyses (GO, KEGG, GSEA, and SCENIC) were performed to assess proliferation, apoptosis, metastasis, and regulatory pathways. In vivo tumorigenesis and metastasis were evaluated in BALB/c nude mice. **Results:** CENPF was significantly upregulated in melanoma tissues and cell lines compared to controls (*p* < 0.05). High CENPF expression correlated with advanced Clark level (*p* = 0.006), ulceration (*p* = 0.04), and poor overall survival (*p* = 0.005). Knockdown of CENPF suppressed melanoma cell proliferation, migration, and invasion in vitro, while inducing G2/M phase arrest and apoptosis. In vivo, CENPF silencing reduced tumor growth and lung metastasis. Mechanistically, CENPF was transcriptionally activated by E2F3, and the E2F3-CENPF axis promoted cell cycle progression via G2/M checkpoint activation and P53 pathway suppression. **Conclusions:** CENPF serves as a prognostic biomarker and therapeutic target in melanoma. Its upregulation drives tumor progression through cell cycle dysregulation and immune evasion, while targeting the E2F3-CENPF axis may offer a novel strategy for melanoma treatment. These findings provide critical insights into melanoma pathogenesis and potential clinical applications.

## 1. Introduction

Melanoma originates from neural crest-derived melanocytes and is the most malignant skin tumor. In recent years, the incidence of melanoma has been on the rise, with an annual growth rate of 3% to 5% [1]. Surgical resection is effective for early melanoma, but melanoma still has the highest mortality rate due to its propensity for abnormal proliferation and early lymphatic and hematogenous metastasis [2]. Once distant metastasis occurs, the prognosis is extremely poor, with a median survival time of only 6 to 9 months and a 5-year survival rate of less than 20% [3]. Despite the advances and breakthroughs in the field of melanoma treatment, an effective diagnostic and therapeutic method is urgently needed. Thus, the mechanisms of proliferation and metastasis of melanoma need to be further studied.

The centromeric protein F (CENPF) is a 350 kDa coiled–coil protein that is transiently expressed in a cell cycle-related pattern. It slowly increases in the cell nucleus during the S phase, and reaches its expression peak in the G2/M phase [4]. At late G2, a sub-pool of CENPF accumulates at the pre-kinetochores. At the onset of anaphase, CENPF translocates from the kinetochores to the central spindle, and later it accumulates at the intracellular bridge between the daughter cells at telophase [5]. At the end of mitosis, CENPF is rapidly proteolyzed [6]. Besides studying its expression changes during the cell cycle, several studies have investigated CENPF’s function during cell mitosis and tumor progression [7,8,9,10,11,12,13]. Huang et al. [7] revealed that CENPF was overexpressed in adrenocortical carcinoma and its increase was associated with dismal survival rates. Moreover, an in vitro experiment showed that knockdown of CENPF arrested cell cycle progression, which manifested as the accumulation of cells in G0/G1 phase, the reduction of cells in G2/M phase, and a slight increase of cells in S phase. However, in another study of hepatocellular carcinoma [8], knockdown of CENPF resulted in the accumulation of cells in G2/M phase and S phase, and the reduction of cells in G0/G1 phase, which indicated that CENPF may function differently in different tumors. In melanoma, a study showed that CENPF was an independent prognostic and metastasis biomarker, and upregulated CENPF might lead to premature depletion of CD4+ memory T cells and immunosuppression [14].

In this study, we re-analyzed three GEO datasets of melanoma and obtained the overlapping differentially expressed genes (DEGs) [15]. Then, the expression levels and prognostic values of these DEGs were detected by Gene Expression Profiling Interactive Analysis (GEPIA) [16]. Among these DEGs, CENPF was selected for additional research. We showed that CENPF was upregulated in melanoma and associated with a poor prognosis. High levels of CENPF were related to a higher Clark level and a greater possibility of ulceration. In vitro and in vivo experiments showed that knockdown of CENPF could inhibit cell proliferation, metastasis and induce apoptosis. Additionally, E2F3 could transcriptionally activate CENPF expression, and silencing CENPF could rescue the cell cycle-promoting effect of the E2F3–CENPF axis. Taken together, our results show that CENPF is a novel prognostic biomarker and potential therapeutic target for melanoma treatment.

## 2. Materials and Methods

### 2.1. Bioinformatics Data Processing

We selected the GSE46517, GSE3189 and GSE7553 profiles from the GEO database (https://www.ncbi.nlm.nih.gov/gds/, accessed on 10 October 2024). GSE46517 included 104 melanoma and 17 normal samples. GSE3189 included 45 melanoma and 25 normal samples (7 normal skin and 18 nevi). GSE7553 included 56 melanoma and 5 nevus samples. DESeq2 (version 1.42.1) was applied to detect differentially expressed genes between the melanoma and normal samples [17] and *p* value < 0.05 and |logFC| ≥ 1 were set as the cutoff criteria. EVenn (http://bioinformatics.psb.ugent.be/webtools/Venn/, accessed on 10 October 2024) is an online tool that can calculate the intersection(s) of lists of elements and was used to obtain the overlapping DEGs in these three databases [15]. GSE75354 included 8 melanoma cell lines and 1 normal melanocyte cell line, and was used to explore the differences in expression of CENPF between melanoma cells and normal melanocytes. GEPIA was used to analyze the gene expression patterns of tumors and control samples and their association with prognosis from the TCGA and GTEx datasets.

### 2.2. GO and KEGG Pathway Enrichment Analysis of DEGs

Pathway analysis of DEGs was conducted using the ClusterProfiler R package (v4.10.1) of R software (v4.3.3) [18] and the top 8 GO functions for biological process, cellular component and molecular function are shown separately in bar plots. The top 10 KEGG pathways are shown in dot plots (if there were 10 or more pathways enriched).

### 2.3. Real-Time Reverse Transcription PCR (qRT-PCR)

Total RNA was extracted using the EZ-Press RNA Purification Kit (EZBioscience, Roseville, CA, USA) and reverse-transcribed using the cDNA Synthesis Kit (Yeasen, Shanghai, China). The primer sequences for CENPF, E2F3 and GAPDH, along with the target sequence used for the knockdown of CENPF and the primer sequences for the luciferase reporter experiments, are detailed in Appendix A. The RNA quantity and quality were verified using a NanoDrop spectrophotometer (Thermo Scientific, Waltham, MA, USA). RT–qPCR was performed using a SYBR-Green Master Mix kit (Yeasen, Shanghai, China) and the conditions were as follows: 95 °C for 5 min, 40 cycles of 95 °C for 10 s plus 60 °C for 30 s, and the melt curve stage was 95 °C for 15 s, 60 °C for 1 min, 95 °C for 1 s. The assays were performed in triplicate using the QuantStudio™ 5 RT-PCR System (Thermo Scientific, Waltham, MA, USA), and the relative gene expression levels were determined using the 2^−ΔΔCt^ method [19].

### 2.4. Western Blot Analysis

For total protein extraction, cell lysates were obtained using RIPA buffer (Beyotime, Shanghai, China) supplemented with phosphatase inhibitors and protease inhibitors. After protein quantification using the BCA Protein Assay Kit (Beyotime, Shanghai, China), equal amounts of protein were resolved by sodium dodecyl sulfate–polyacrylamide gel electrophoresis (SDS–PAGE) (Epizyme, Shanghai, China) and transferred onto polyvinylidene fluoride (PVDF) membranes (Merck Millipore, Burlington, VT, USA). After blocking with QuickBlock™ Blocking Buffer (Beyotime, Shanghai, China) for 15 min, the membranes were incubated overnight at 4 °C with primary antibody. After washing with Tris-buffered saline with Tween 20 (Sangon Biotech, Shanghai, China), the membrane was probed with HRP-conjugated secondary antibody (Cell Signaling Technology, Danvers, MA, USA), and the signals were detected using ECL detection reagents (Beyotime, Shanghai, China). The antibodies used in this study are listed in Appendix A.

### 2.5. Tissue Microarray (TMA) Construction and Immunohistochemical Staining

A total of 128 paraffin-embedded melanoma and paired peritumor tissue microarrays were analyzed by immunohistochemistry (IHC). All patients underwent complete excision, and tissue specimens were verified by pathological examination. The patients did not receive radiotherapy or chemotherapy before the operation. The clinical stage of the patients was determined by tumor–node–metastasis staging according to the American Joint Committee on Cancer classification system (8th edition) [20]. Ethics approval was obtained from the Ethics Committee of the Zhongshan Hospital Biomedical Research Department, and all patients signed informed consent forms. For IHC, tissue slides were deparaffinized and rehydrated using a gradient concentration of xylene and ethanol. After incubation with 0.3% hydrogen peroxide for 30 min, antigen retrieval was performed with citrate buffer at a sub-boiling temperature for 15 min, and then blocked with 5% bovine serum albumin for 60 min. The slides were incubated with the primary antibody (anti-CENPF, 1:500 dilution, ab5, ABCAM) overnight at 4 °C and then incubated with HRP-conjugated secondary antibodies at 37 °C for 60 min. Slides were then stained with diaminobenzidine (Beyotime, Shanghai, China). Images were obtained using the CaseViewer software (Version 2.4) (3DHISTECH, Budapest, Hungary). Immunoreactivity was graded as previously described [21]. Briefly, the score was determined based on a combination of percent staining (1 [0–25%], 2 [>25–50%], 3 [>50–75%], or 4 [>75–100%]) and staining intensity (0 [negative], 1 [low], 2 [moderate], or 3 [high]). The sum of the 2 scores for each specimen was taken as the expression level of CENPF, which was categorized as low (0–3) or high (4–7).

### 2.6. Function Exploration of CENPF in Melanoma and Gene-Set Enrichment Analysis (GSEA)

The function of CENPF was explored based on the TCGA SKCM dataset, of which the top 10% samples (47 patients) that had the highest expression of CENPF were categorized as CENPF-high and the lowest 10% (47 patients) were categorized as CENPF-low. DEGs between the two groups were found using DESeq2. Then, GO function and KEGG pathway enrichment analyses were carried out as described above. GSEA of the DEGs also used the ClusterProfiler R package. The predefined gene set was ‘h.all.v2023.2.Hs.symbols.gmt’ [22]. Normalized enrichment scores (NES) were reckoned as the main GSEA statistical results. Statistical significance thresholds were set as |NES| > 1, normalized *p*-values (NOM *p*-values) < 0.05 and FDR < 0.25.

### 2.7. Exploration of CENPF Expression and Immune Properties

The estimate R package (v1.0.13) was used to explore the relationship between CENPF expression and immune infiltration of tumor microenvironment [23]. The CIBERSORT R package (v0.1.0) was used to infer the abundance of 22 immune cells in 94 TCGA SKCM patients [24]. Then, a lollipop plot was used to show the relationship of 22 immune cells’ abundance with the expression of CENPF. The TIDE website (http://tide.dfci.harvard.edu/, accessed on 10 October 2024) was used to predict the response to immunotherapy of the two groups [25]. Similarly, the Oncopredict R package (v1.2) was used to predict the sensitivity of the two groups to multiple chemotherapy drugs [26].

### 2.8. Cell Culture, Plasmid Construction and Transfection

The melanoma cell lines, including SK-MEL-28, A2058, SK-MEL-2, MV3 and A375, were purchased from the Cell Bank of the Chinese Academy of Sciences (Shanghai, China) and they were grown in high-glucose DMEM medium (GENOM, Nanjing, China). Knockdown of CENPF was performed using specific shRNAs delivered by a lentiviral system, purchased from Shanghai OBiO Tech, Inc. (Shanghai, China), according to the manufacturer’s instructions with pLenti-shRNA-NC used as the negative control. Subsequently, the cells with suitable fluorescent expression were selected with puromycin at a concentration of 2 µg/mL. The transfection efficiency was verified by RT–qPCR and Western blotting. Plasmids used for the luciferase reporter assay were purchased from Shanghai Genechem Co., Ltd. (Shanghai, China). Knockdown of E2F3 was performed using siRNA purchased from Tsingke Biotechnology Co., Ltd. (Shanghai, China).

### 2.9. Wound Healing and Transwell Assays

For the wound healing assay, when cells reached 99% confluence, a 1000 μL pipette tip was used to scratch the bottom of the plate; serum-free DMEM was added, and the cells were photographed and then cultured in an incubator at 37 °C and 5% CO_2_. The cells were photographed again at 16 h (SK-MEL-28) or 24 h (A2058), and the migration rate was calculated using ImageJ (Version 1.54f) software (National Institutes of Health, Bethesda, MD, USA). For the Transwell migration assay, 2 × 10^4^ cells with FBS-free medium were seeded into the upper chamber of 8.0 μm pore Transwells (Corning, Corning, NY, USA); medium containing 10% fetal bovine serum was added to the lower chamber. After 48 h, the chamber was removed and washed with PBS, then fixed in 4% paraformaldehyde solution for 30 min and stained with crystal violet for 15 min. The samples were photographed under a microscope.

### 2.10. EdU, CCK-8, and Colony Formation Assays

For the EdU assay, cells at 50% confluency were cultured with EdU labeling medium (Beyotime, Shanghai, China) for 2 h, fixed with 4% paraformaldehyde for 30 min and treated with Immunol Staining Wash Buffer (Beyotime, Shanghai, China) containing Triton X-100 for 15 min at room temperature. After 3 washes with QuickBlock™ Blocking Buffer for Immunol Staining (Beyotime, Shanghai, China), the cells were dyed with Azide 555 (Beyotime, Shanghai, China) for 30 min. Then, Hoechst (Beyotime, Shanghai, China) was used to stain the DNA in the cells, and subsequently they were visualized with a microscope. For the CCK8 assay, cells were inoculated into 96-well plates (1000 cells/well). In each sample, the original medium was replaced by medium with 100 μL 10% CCK-8 reagents (Beyotime, Shanghai, China) at different time points (24, 48, 72, and 96 h), followed by 2 h incubation at 37 °C. The absorbance of each sample was then measured at 450 nm. For the colony formation assay, cells were seeded in a six-well plate (1000 cells/well for SK-MEL-28, 500 cells/well for A2058) with the culture medium refreshed every 3 days for 2 weeks. Following the 2-week period, the cells were washed with PBS, fixed with 4% paraformaldehyde and stained with 0.4% crystal violet for 15 min. The number of colonies containing more than 10 cells was counted manually and averaged over duplicate wells.

### 2.11. Flow Cytometry for Cell Cycle and Cell Apoptosis

For cell cycle detection, A375 and SK-MEL-28 cells were harvested when they reached approximately 90% confluence. The cells were washed once with pre-cooled PBS (4 °C) and then fixed with pre-cooled 75% ethanol at 4 °C overnight. After washing with PBS, the cells were resuspended in the cell staining solution from a Cell Cycle and Apoptosis Analysis Kit (Beyotime, Shanghai, China), and then a flow cytometer (Beckman Coulter, Brea, CA, USA) was used to detect the cell cycle distribution. Cell apoptosis detection was performed using an Annexin V-APC/PI Apoptosis Kit (Multi Sciences, Hangzhou, China). According to the protocol, 1 × 10^6^ cells were washed twice with PBS and induced to death with 500 μL Apoptosis Positive Control Solution. Then the dead cells were mixed with live cells. After staining with APC and PI colorants from the kit, they were used for adjusting the fluorescence channel voltage. Then, the experimental group cells and control group cells were stained and analyzed using the adjusted voltage. All the flow cytometry data were analyzed using FlowJo10.8.1 software.

### 2.12. In Vivo Tumorigenesis and Metastasis

Xenograft experiments in nude mice were approved by the Animal Experimentation Ethics Committee of Zhongshan Hospital, Fudan University. The healthy 6-week-old male BALB/c nude mice were randomly divided into four subgroups (n = 6/each group): one control group (Control) and one treatment group (sh CENPF) for establishing subcutaneous xenograft tumors and one control group (Control) and one treatment group (sh CENPF) for establishing lung metastasis models. A total of 5 × 10^6^cells (per mouse) were injected to establish subcutaneous xenograft tumor models as previously described [21]. In addition, 1 × 10^6^ cells (per mouse) were injected through the tail vein to establish the lung metastasis models. Tumor growth was monitored every 5 days, and the mice were sacrificed after 30 days. The tumor and lung tissues from different groups were fixed in formalin and embedded in paraffin. Consecutive sections were prepared for each tissue block and subcutaneous xenograft tumors were stained with H&E and Ki67 while the lung tissues were only stained with H&E. The tumor volume was measured using the following formula: V = π/6 × (larger diameter) × (smaller diameter)^2^. The presence of lung metastases was calculated and evaluated independently by two pathologists.

### 2.13. Transcription Factors Inference of CENPF

SCENIC R package (v 1.3.1) was used to infer the transcription factors of CENPF using 94 patients’ RNA expression matrices from TCGA [27]. In detail, we identified transcription factor binding motifs over-represented on a gene list with the RcisTarget package (v1.18.2). The activity of each group of regulons in each cell was scored by the AUCell package (v1.24.0). Then, the transcription factors inferred by SCENIC were re-checked using the JASPAR database and a threshold of 75%. Finally, based on the hypothesis that this transcription factor’s RNA expression is elevated and is associated with a poor prognosis, we identified E2F3 as a potential transcription factor.

### 2.14. Luciferase Reporter Assay

For the detection of the binding site of E2F3 on the CENPF promoter, the indicated fragments of the CENPF promoter or negative control fragments were cloned into the GV534 vector and transfected into SK-MEL-28 and A2058 cells. Luciferase activity was detected using the dual-luciferase reporter assay system (Yeasen, Shanghai, China) according to the manufacturer’s instructions. Briefly, the expression vectors or control vectors were co-transfected with the reporter plasmids into cells using Lipofectamine 3000 reagent (Thermo Scientific, Waltham, MA, USA). At 48 h post transfection, the cells were lysed and the luciferase substrate was added to the cell lysate. The luciferase activity was measured using Softmax Pro 7 (Molecular Devices, San Jose, CA, USA), and firefly luciferase activity was normalized to that of Renilla luciferase.

### 2.15. Statistical Analysis

All experiments were performed in triplicate. All data are presented as means ± standard deviations (SDs). Kaplan–Meier analysis and the log-rank test were used to compare survival between different groups. The Shapiro–Wilk normality test was first used to test the data distribution for continuous variables. For normally distributed variables, statistical significance was determined using the t-test for two-group comparisons and two-way ANOVA for multiple-group comparisons. For non-normally distributed variables, two-group comparisons were conducted using the Mann–Whitney U test, and multiple-group comparisons were performed using the Kruskal–Wallis test. Correlations between two variables were determined using Pearson’s correlation coefficient. Categorical data were analyzed using the chi-square test or Fisher’s exact test. Cox proportional hazards regression was used to analyze independent prognostic factors. A *p*-value of <0.05 was considered statistically significant. In the figures, we use different numbers of stars to represent different levels of significance: “*” indicates *p* < 0.05; “**” indicates *p* < 0.01; “***” indicates *p* < 0.001; “****” indicates *p* < 0.0001.

## 3. Results

### 3.1. Identification of Key Genes in Melanoma Using Public Datasets

In order to find the oncogenes that facilitate melanoma’s progression, we analyzed three public GEO datasets. GSE46517 contained 104 melanoma and 17 normal skin samples, GSE3189 contained 45 melanoma and 25 normal skin samples, and GSE7553 contained 56 melanoma and 5 normal skin samples. After performing differential expression analysis, we identified 699 DEGs (275 upregulated and 424 downregulated) from GSE46517, 1918 DEGs (694 upregulated and 1224 downregulated) from GSE3189, and 2580 DEGs (1462 upregulated and 1118 downregulated) from GSE7553 (Figure 1A). In order to find key target genes that exert crucial roles in melanoma, we took an intersection of the upregulated DEGs in these datasets (Figure 1B). We also conducted GO and KEGG analyses to explore the enriched functions and pathways in melanoma. Both GO and KEGG showed that 93 intersecting genes of GSE46517 and GSE3189 were enriched in cell division and the cell cycle (Figure 1C). GO analysis of 149 intersecting genes of GSE7553 and GSE3189 showed similar results, which was consistent with the hyperproliferative property of melanoma. For further verification, we submitted the 54 central upregulated genes to GEPIA to detect the differences in gene expression levels between the tumor and normal samples and the association of these DEGs with patients’ outcomes. We found that CENPF, E2F3, KIRREL, METRN were both upregulated and associated with worse OS in melanoma patients (Figure 1D,E). Our previous study [28] using scRNA-seq data revealed a subset of melanoma cells with high proliferative ability, with CENPF being the top differential gene. All of these lines of evidence indicate that CENPF may play an important role in melanoma progression.

### 3.2. The Expression of CENPF Was Significantly Upregulated in Melanoma and Was Associated with a Worse Prognosis

We explored the clinical significance of CENPF in melanoma. Firstly, the violin plots showed CENPF was highly expressed in melanoma in the above three GEO datasets (Figure 2A). Moreover, the GSE75356 profile, which contained the RNA sequencing data of eight melanoma cell lines and one human normal melanocyte line, showed that CENPF was significantly upregulated in all the melanoma cell lines (Figure 2B). Similarly, in our clinical specimens, CENPF’s mRNA level was found to be increased in 11/12 melanoma tissues compared with peritumor tissues (Figure 2C). Western blot analysis also showed that the expression level of the CENPF protein was higher in melanoma tissues than that in adjacent tumor tissues, and representative images are shown in Figure 2D. In addition, we investigated the expression of CENPF in a TMA containing 128 melanoma tissue samples and found CENPF was indeed upregulated in most of the tumors compared with peritumor tissues (Figure 2F). 

We observed varying levels of CENPF protein expression in tumor tissues (−, absent; +, weak; ++, moderate; +++, strong) and representative images are shown in Figure 2G. Then, the absent and weak samples were categorized as CENPF-low, while moderate and strong samples were categorized as CENPF-high, resulting in 76 patients with low expression and 52 patients with high expression. The relationship between the expression of CENPF and the clinical information of melanoma patients is displayed in Table 1. We found that the CENPF-high group had a lower overall survival (OS) rate (*p* = 0.005, Figure 2H) and a lower disease-free survival (DFS) rate (*p* = 0.01) than the CENPF-low group. In addition, higher CENPF expression was associated with a higher Clark level (*p* = 0.006) and a greater possibility of ulceration (*p* = 0.04). Univariate and multivariate analyses showed that Clark level, lymphatic metastasis, distant metastasis, clinical stage and CENPF staining were significantly associated with OS and DFS (Table 2; the forest plots are shown below). However, in multivariate analysis, only the CENPF expression level was a significant independent prognostic variable for the prediction of OS. In conclusion, CENPF was significantly upregulated in melanoma and was associated with a worse prognosis in melanoma.

### 3.3. CENPF Facilitated Melanoma Progression Through Promoting Cell Proliferation, Inhibiting Cell Apoptosis and Influencing Melanoma’s Immune Properties

In order to elucidate the potential pathway of CENPF’s function in melanoma, we further analyzed the TCGA SKCM datasets, in which the 10% of samples that had the lowest CENPF expression were categorized as CENPF-low and the highest 10% were categorized as CENPF-high. Figure 3A exhibits the expression of CENPF and other clinical parameters. Then, we carried out GO function and KEGG pathway enrichment analyses (Figure 3B). GO function analysis showed that upregulated genes were enriched in chromosome segregation, nuclear chromosome segregation, nuclear division and meiotic cell cycle and so on, which is actually in accordance with the function of CENPF as its main role in forming the centromere and interacting with the kinetochore. Meanwhile, the downregulated genes were enriched in epidermis development, skin development, and keratinocyte differentiation and so on. KEGG pathway enrichment analysis showed that the upregulated genes were enriched in the cell cycle, calcium signaling pathway and motor proteins, while the downregulated genes were enriched in *Staphylococcus aureus* infection, arachidonic acid metabolism and viral protein interactions with cytokines and cytokine receptors and so on. Moreover, we also used GSEA to investigate the cancer-related signaling pathways associated with CENPF’s expression (Figure 3C). We found that CENPF-high samples had activated E2F targets, G2M checkpoint, mitotic spindle and myc targets, while they had a suppressed P53 pathway. 

We also analyzed CENPF’s expression with immune properties. Firstly, ESTIMATE showed CENPF-high samples had lower immune scores (Figure 3D). CIBERSORT showed that several immune cells were differently enriched between the two groups (Figure 3E), such as CD8+ T cells and activated NK cells (Figure 3F), which may result in the lower immune state of CENPF-low patients. Finally, we used the TIDE website and Oncopredict R package to predict patients’ responses to drug therapy and found that CENPF-high patients had a lower response to immunotherapy and less sensitivity to most chemotherapy drugs (Figure 3G and Appendix A). In general, these results suggested that CENPF facilitated melanoma progression through promoting cell proliferation, inhibiting cell apoptosis and influencing melanoma’s immune properties.

### 3.4. The Knockdown of CENPF Inhibited the Proliferation and Metastasis of Melanoma Cells In Vitro

Then, we conducted cell experiments to verify the above analysis. Firstly, we detected the expression of CENPF in five melanoma cell lines (SK-MEL-28, A2058, SK-MEL-2, MV3, and A375) by qRT-PCR (Figure 4A) and chose SK-MEL-28 and A2058 cells, which had the highest expression of CENPF for further experiments. We then knocked down CENPF by shRNAs and verified the downregulation by qRT-PCR (Figure 4B) and Western blot (Figure 4C). After the downregulation of CENPF, cell proliferation was indeed decreased as assessed by EdU, CCK-8 and colony formation assays (Figure 4D–F). Moreover, although the bioinformatic analysis did not show a direct relationship between CENPF and cell metastasis, previous studies in other cancers indicated that downregulation of CENPF inhibited cell metastasis [7,13]. Therefore, we also measured the cell metastatic ability by wound healing and Transwell assays and found that downregulation of CENPF significantly inhibited cell metastasis (Appendix A). Taken together, these results indicated that CENPF could promote the proliferation and inhibit metastasis of melanoma cells in vitro.

### 3.5. The Knockdown of CENPF Arrested Melanoma Cells in G2/M Phase and Increased Cell Apoptosis

To investigate the mechanism of CENPF in melanoma proliferation, we performed flow cytometric analysis. The results showed that downregulation of CENPF increased the proportion of cells in the G2/M and S phases while decreasing the proportion of cells in the G0/G1 phase (Figure 5A). This finding was consistent with the GSEA results, which indicated that CENPF’s function was positively enriched in the G2M checkpoint. Western blot assays showed that the G1 phase marker cyclin D1 was downregulated, S phase markers CDK2 and cyclin E1 were increased, and G2/M phase markers CDK1 and cyclin B1 were downregulated after knockdown of CENPF (Figure 5B), which indicated that melanoma cells transitioned more slowly to G2/M phase, with some being arrested in the G2/M phase. Moreover, we also verified CENPF’s negative function in the P53 pathway by flow cytometry and found CENPF’s downregulation indeed increased cell apoptosis (Figure 5C). Western blot assays showed an increase of P53, P21, and Bax, which participate in the P53 pathway (Figure 5D). In conclusion, these results indicated that knockdown of CENPF arrested melanoma cells in G2/M phase and increased cell apoptosis.

### 3.6. The Knockdown of CENPF Inhibits Melanoma Growth and Metastasis In Vivo

To further confirm the effect of CENPF in vivo, stable CENPF-knockdown SK-MEL-28 cells and negative control cells were subcutaneously injected into nude mice. Thirty days after tumor implantation, we observed that CENPF knockdown significantly inhibited tumor growth as measured by tumor volume and weight (Figure 6A–C). IHC showed that tumors from CENPF-knockdown cells had reduced Ki67 expression, which represented lower proliferative capacity (Figure 6D). Moreover, we also established a lung metastasis model through tail vein injection. As shown in Figure 6E, the CENPF downregulation group had fewer lung metastasis foci. In summary, these results indicated that CENPF knockdown could inhibit melanoma growth and metastasis in vivo.

### 3.7. Upregulation of CENPF Was Regulated by E2F3 in Melanoma Cells

Based on the promoting effect of CENPF in melanoma development, we further explored the mechanisms of CENPF’s upregulation in melanoma cells. We conducted a SCENIC analysis, which can predict the potential transcription factors of CENPF using the TCGA SKCM datasets as mentioned in part 2 (Figure 7A, Appendix A). The results identified several transcription factors such as BCLAF1, MYBL1 and TAF1 and so on. We then intersected the SCENIC results with the predicted results from the JASPAR database, using a 75% threshold, and found E2F2, E2F3, E2F6, E2F7 and MYBL1 (Appendix A). We then checked their expression and prognostic correlations in melanoma samples using TCGA SKCM datasets and GTEx normal samples and found that only E2F3 was both upregulated in melanoma and associated with a poor prognosis (Figure 1E, Appendix A). At the same time, the TCGA SKCM dataset showed that E2F3 was positively correlated with CENPF (R = 0.41, *p* < 0.001, Figure 7B). Moreover, E2F3 was reported to promote melanoma cell growth through copy number variation [29]. Other studies reported that a decrease of several non-coding RNAs resulted in an increase of E2F3 in melanoma [30,31,32,33]. Both qPCR and Western blot assays showed that the upregulation of E2F3 indeed increased the expression of CENPF in two cell lines, but CENPF knockdown did not influence E2F3 expression (Figure 7C,D). Luciferase reporter gene assays verified that E2F3 could transcriptionally activate CENPF (Figure 7E). The above results showed that E2F3 could transcriptionally activate CENPF expression in melanoma cells.

### 3.8. Rescue Experiments Verified the Effect of Silencing CENPF in Reversing the E2F3–CENPF Axis

Firstly, we over-expressed E2F3 alone or both overexpressed E2F3 and knocked down CENPF in SK-MEL-28 cells. Previous studies confirmed that knockdown of E2F3 could increase G1 phase and S phase cells, while decreasing G2 phase cells [29]. Western blot assays confirmed that over-expressed E2F3 would decrease G1 phase proteins such as cyclin D1 and increase G2/M phase proteins such as cyclin B1 and CDK1 (Figure 8A). Both overexpressing E2F3 and knocking down CENPF reversed the cell cycle-promoting effect of E2F3 and arrested cells in G2/M phase. This was reflected by a decrease in G1 phase proteins and G2/M phase proteins, along with an increase in apoptosis proteins (Figure 8A). CCK8 assays also showed the impaired proliferative ability following CENPF knockdown (Figure 8B). In conclusion, we identified the E2F3–CENPF axis in melanoma, which contributes to melanoma’s high proliferative ability, and demonstrated that knocking down CENPF could inhibit melanoma cell proliferation and induce cell apoptosis (Figure 8C).

## 4. Discussion

As sequencing technology advances, more and more sequencing data have been generated, but most of these data are analyzed only once. We re-analyzed three GEO datasets, intersected the upregulated genes, and identified four genes that may be associated with melanoma’s malignant phenotype. Several studies have reported that these genes were associated with melanoma. For example, Zhicai Feng et al. [29] found that copy number amplification and other mechanisms result in high expression of E2F3 in melanoma, which promotes tumor progression by involving the cell cycle. Liron Zehavi et al. [34] revealed that the down regulation of miR-377 also resulted in the upregulation of E2F3 and the MAP3K7/NF-kB signaling pathway in melanoma cells. Sebastian Lundgren et al. [35] verified that KIRREL was upregulated in melanoma and high KIRREL protein expression was an independent factor correlating with recurrence-free and melanoma-specific survival, particularly in thin melanomas, even outperforming absolute thickness and ulceration. Mengzhi Li et al. [14] found that CENPF was an independent prognostic and metastasis biomarker associated with CD4+ memory T cells in cutaneous melanoma. And Haiting Xu et al. [36] showed that miR-383-5p expression was downregulated in melanoma cell lines, which resulted in the increase of CENPF. Besides, in our previous study [28], single-cell RNA sequencing revealed a subset of melanoma cells of high proliferation ability, with CENPF being the top differentially expressed gene. Therefore, we further aimed to determine the role of CENPF in melanoma’s high proliferation ability.

Firstly, we verified that CENPF mRNA and protein expression was upregulated in melanoma tissues compared to normal tissues, and patients with a high level of CENPF had a poorer prognosis than those with a low level of CENPF. Clinically, CENPF-high patients had a higher Clark level and a greater possibility of ulceration. Univariate analyses showed that CENPF could be an independent prognostic factor for both OS and DFS in melanoma patients. However, in multivariate analyses, the predictive effect was only statistically significant for OS. Using TCGA datasets, we conducted GO, KEGG and GSEA analyses and found CENPF’s function was positively enriched in E2F targets and the G2M checkpoint and negatively enriched in the P53 pathway, which actually aligned with CENPF’s known function. As a centromere protein, CENPF interacts with the kinetochore and helps microtubules to attach to chromosomes, enabling the chromosomes to be pulled to the poles of the cells so that the cells could successfully progress through G2/M phase [4,5,6]. Dysfunction in chromosome assignment may result in cell apoptosis, which explains why CENPF is negatively associated with the P53 pathway. 

In addition, considering that a previous study showed CENPF was negatively related to CD4+ memory T cells, we also re-explored the correlation between CENPF and immune properties and found CD8+ T cells and activated NK cells were also negatively associated with CENPF expression, which may explain the worse immune response predicted by ESTIMATE and TIDE. Then, we knocked down CENPF in SK-MEL-28 and A2058 cells and found this indeed impaired cell proliferation. Next, flow cytometric analysis was performed, which showed that silencing CENPF increased the percentage of cells in G2/M phase and S phase but decreased the percentage of cells in G0/G1 phase, which indicated that cells were arrested in G2/M phase. Similar to our results, Penghui Xu et al. [13] showed that CENPF deficiency arrested the cell cycle in the G2 phase and decreased the expression of G2 phase markers (CDK1 and cyclin B1) in human gastric cancer cells. Yongdong Dai et al. [8] showed a similar result in liver cancer cells. However, in adrenocortical carcinoma cells, Yu-gang Huang et al. [7] found that siCENPF impaired cell proliferation associated with an accumulation of cells in G0/G1 phase, suggesting that CENPF may have different functions or may function earlier in human adrenocortical carcinoma. 

Moreover, the flow cytometric analysis confirmed that knockdown of CENPF resulted in increased cell apoptosis. Western blot assays showed decreased expression of G2/M phase markers CDK1 and cyclin B1 and G0/G1 phase marker cyclin D1, but increased expression of S phase marker CDK2 and cyclin E1 after silencing CENPF. In vivo assays also showed CENPF downregulation significantly weakened tumor growth and metastasis and that CENPF expression was positively correlated in tumor tissue with Ki67, which is a widely accepted biomarker for cell proliferation. Considering that CENPF may function more as a downstream protein involved in cell division, we further explored potential transcription factors that contribute to its upregulation. By combining the prediction results from SCENIC and the JASPAR database, the expression differences between tumor and normal tissues, and correlating these with prognosis, we finally deduced E2F3 may be the key transcription factor and verified this by luciferase reporter gene assays. Q-PCR assays and rescue experiments further confirmed the regulatory role of E2F3 in CENPF expression. Silencing CENPF inhibited the cell cycle-promoting effect of the E2F3–CENPF axis. From all the above results, it can be concluded that CENPF plays an important role in melanoma progression and may serve as a promising target for melanoma treatment.

## 5. Conclusions

In conclusion, the results of this study show the important role of CENPF in promoting melanoma proliferation and inhibiting cell apoptosis via E2F targets, the G2M checkpoint and the P53 pathway. Knockdown of CENPF arrests cells in G2/M phase and increases cell apoptosis, which reverses the cell cycle-promoting effect of the E2F3–CENPF axis. These findings provide new insight into the molecular basis of melanoma progression and suggest that CENPF is a useful prognostic indicator and potential therapeutic target for melanoma treatment.

## Figures and Tables

**Figure 1 biomedicines-13-00792-f001:**
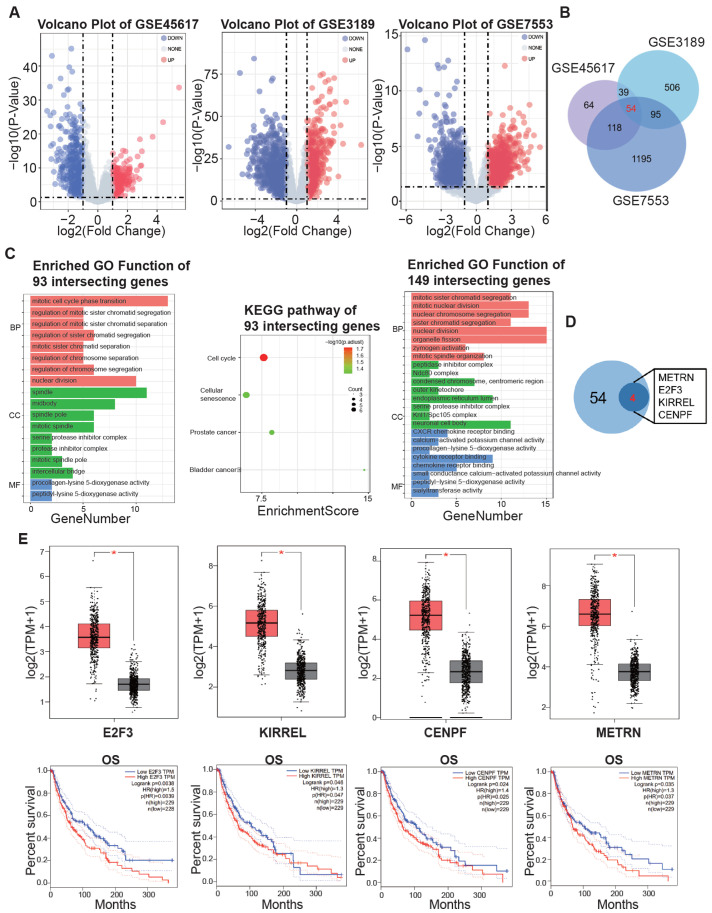
**Identification of key genes in melanoma using public datasets.** (**A**) Volcano plots of the differentially expressed genes in GSE46517, GSE3189 and GSE7553. (**B**) Venn diagram showing the number of intersecting genes in the three GEO datasets. (**C**) Representative images of enrichment analysis of the intersecting genes in at least two datasets. Bar plot of GO functions and dot plot of KEGG pathways enriched from the 93 intersecting genes of GSE46517 and GSE3189 (left and middle). Bar plot of GO functions enriched from the 149 intersecting genes of GSE3189 and GSE7553 (right). (**D**) Venn diagram showing the four genes both upregulated and associated with worse overall survival in melanoma patients. (**E**) Relative expression of the four genes from the GEPIA analysis (upper panel, red box = tumor, black box = normal). The red asterisk indicates a significant difference between the two groups. Kaplan–Meier curves showed the correlation of overall survival with the four genes (lower panel).

**Figure 2 biomedicines-13-00792-f002:**
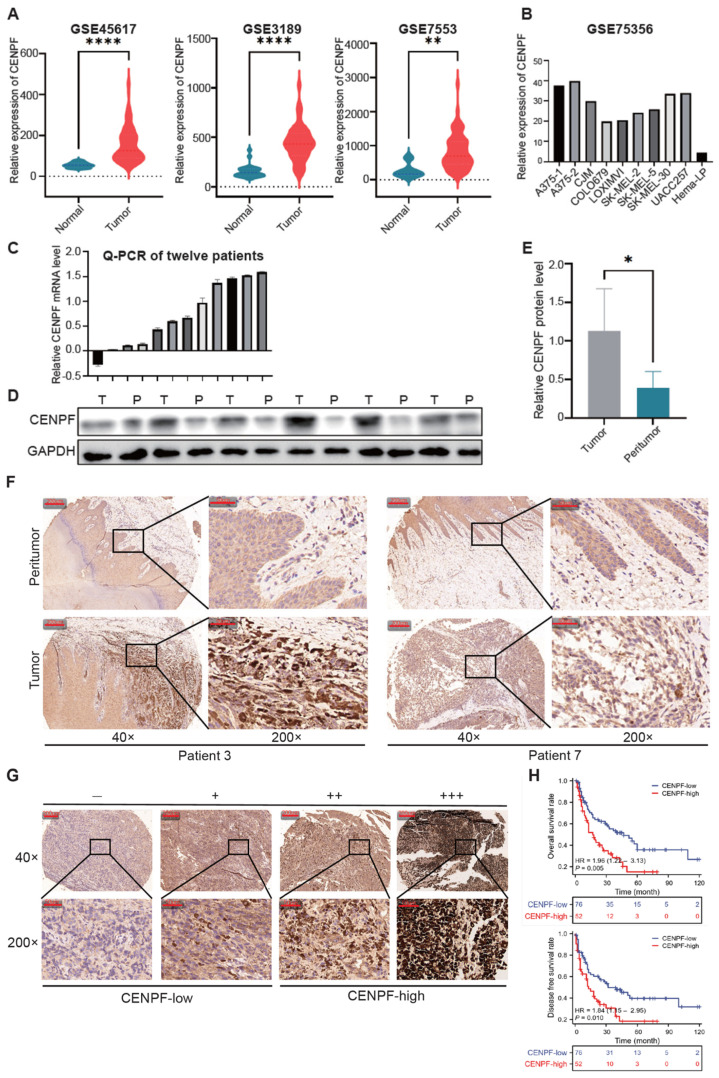
**The expression of CENPF was significantly upregulated in melanoma and was associated with a worse prognosis.** (**A**) Violin plots showing the expression of CENPF in normal samples and tumor patients in the GSE46517, GSE3189 and GSE7553 datasets. “*” indicates *p* < 0.05; “**” indicates *p* < 0.01; “****” indicates *p* < 0.000, and same for all figures below. (**B**) Bar plot showing the expression of CENPF in eight melanoma cell lines and one normal melanocyte cell line in GSE75356. (**C**) CENPF mRNA expression in 12 pairs of melanoma tissues shown as Log2 (T/P). T tumor, P peritumor. (**D**,**E**) CENPF protein expression in 12 pairs of melanoma tissues; representative bands are shown. (**F**) Representative images of the TMA stained with IHC for CENPF. (**G**) Representative images of tumor tissues in different staining classifications are shown and graded from “−” to “+++”. (**H**) Kaplan–Meier curves showing the relationship between CENPF expression and OS and DFS.

**Figure 3 biomedicines-13-00792-f003:**
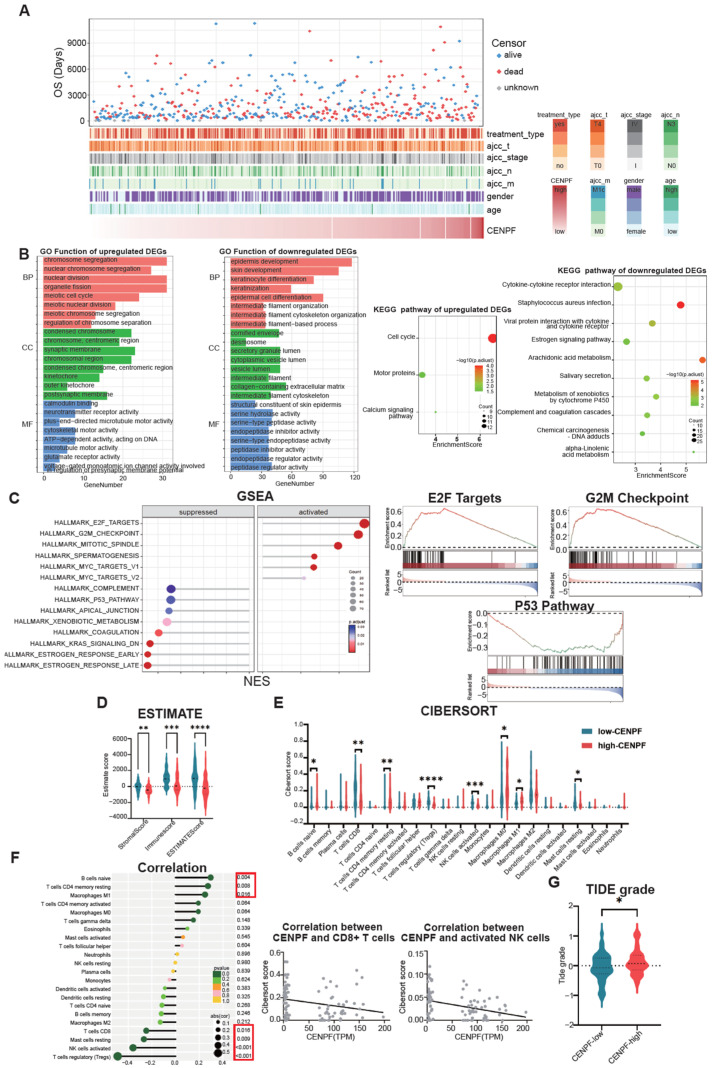
CENPF facilitates melanoma progression through promoting cell proliferation, inhibiting cell apoptosis and influencing melanoma’s immune properties. (**A**) Landscape showing the relationship between CENPF expression and clinical features. (**B**) Bar plots and dot plots showing the GO functions and KEGG pathways enriched from the DEGs between CENPF-low and CENPF-high patients. (**C**) GSEA analysis of Hallmark pathways indicating that high CENPF is positively related to the cell cycle and negatively related to cell apoptosis (left panel). Representative graphs showing the enrichment results of E2F target, G2M checkpoint and P53 pathway (right panel). (**D**) Violin plot showing the ESTIMATE score of CENPF-low and CENPF-high patients. (**E**) Violin plot showing the CIBERSORT results of the abundance of 22 immune cells. (**F**) Lollipop chart showing the correlation between CENPF expression and the abundance of 22 immune cells (left panel). Dot plots showing the correlation of CENPF expression with CD8+ T cells and activated NK cells (right panel). (**G**) Violin plot showing the TIDE scores of CENPF-low and CENPF-high patients. “*” indicates *p* < 0.05; “**” indicates *p* < 0.01; “***” indicates *p* < 0.001; “****” indicates *p* < 0.0001.

**Figure 4 biomedicines-13-00792-f004:**
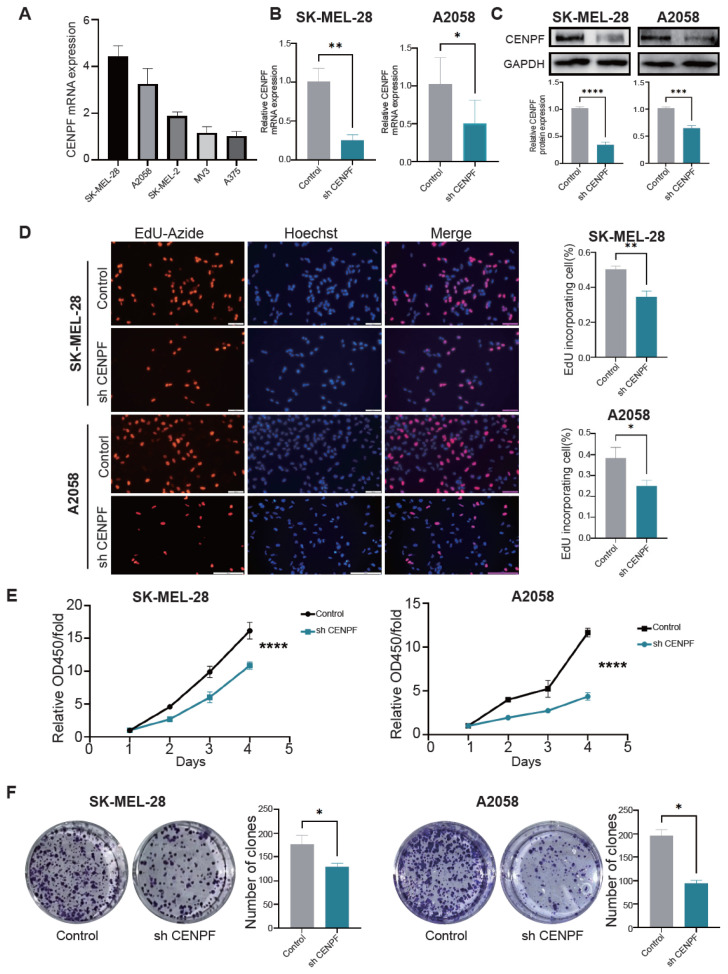
**Knockdown of CENPF inhibited cell proliferation and migration in vitro.** (**A**) qRT-PCR assay showing the expression level of CENPF in five melanoma cell lines. (**B**,**C**) qRT-PCR and Western blot assays confirming the successful knockdown of CENPF in SK-MEL-28 and A2058 cells. (**D**–**F**) EdU (**D**), CCK-8 (**E**) and colony formation (**F**) assays assessing the proliferation of SK-MEL-28 and A2058 cells. “*” indicates *p* < 0.05; “**” indicates *p* < 0.01; “***” indicates *p* < 0.001; “****” indicates *p* < 0.0001.

**Figure 5 biomedicines-13-00792-f005:**
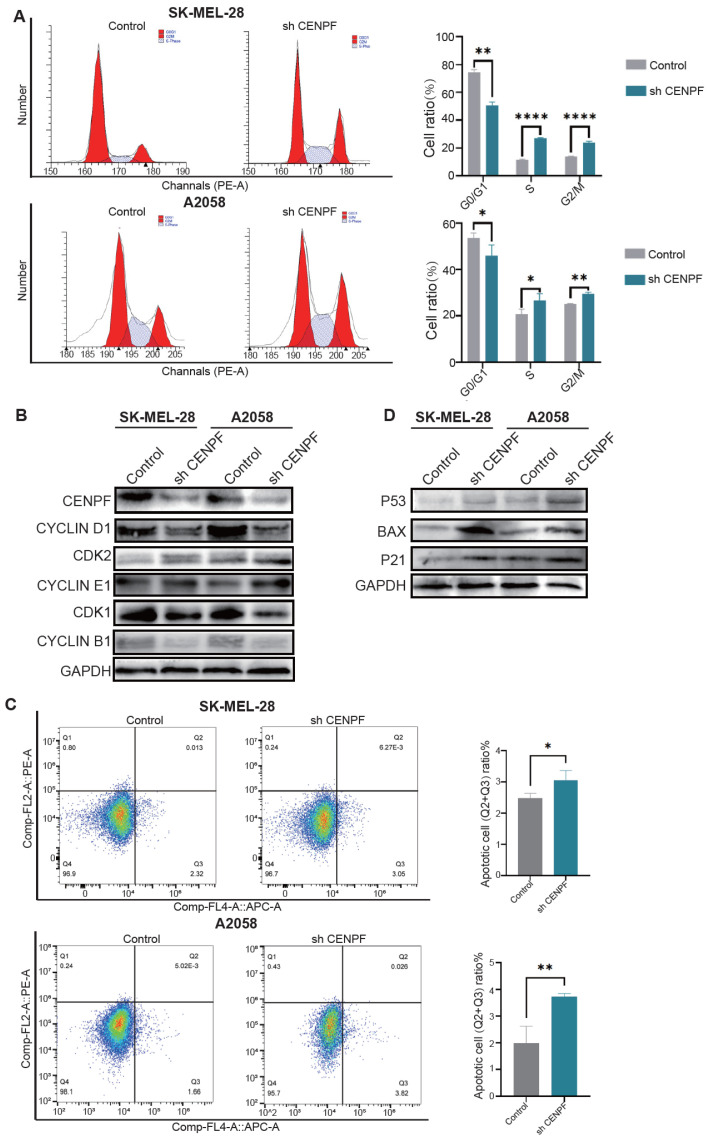
**Knockdown of CENPF arrested melanoma cells in G2/M phase and increased cell apoptosis.** (**A**) Flow cytometry analysis showing an increase in cells in the G2/M phase and S phase and a decrease in cells in the G0/G1 phase after knocking down CENPF. (**B**) Western blot assays confirming the decrease of G2/M markers CDK1 and cyclin B1 and G1 phase marker cyclin D1 and the increase of S phase markers CDK2 and cyclin E1 after knocking down CENPF. (**C**) Flow cytometry analysis showing the increase of cell apoptosis after knocking down CENPF. (**D**) Western blot assays confirming an increase of proteins associated with the P53 pathway after knocking down CENPF. “*” indicates *p* < 0.05; “**” indicates *p* < 0.01; “****” indicates *p* < 0.0001.

**Figure 6 biomedicines-13-00792-f006:**
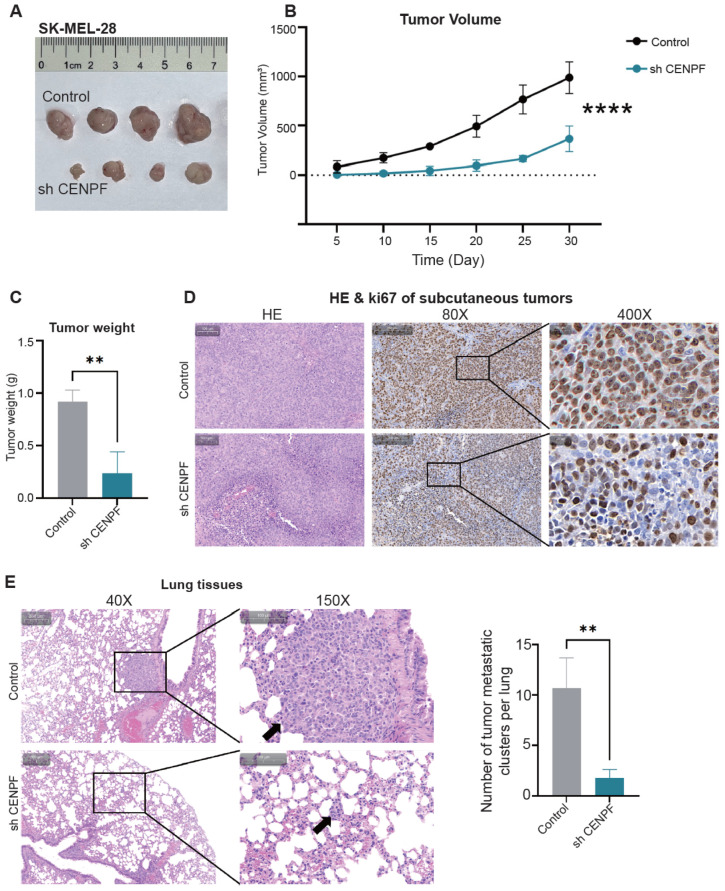
**Knockdown of CENPF inhibits melanoma growth and metastasis in vivo.** (**A**) The subcutaneous xenograft tumors were collected on day 30. (**B**) Subcutaneous xenograft tumor volumes were measured at the indicated time points. (**C**) Bar plot showing the weights of subcutaneous xenograft tumors. (**D**) Subcutaneous xenograft tumors were stained with H&E and IHC for Ki67. (**E**) Lung tissues stained with H&E; representative images are shown. “**” indicates *p* < 0.01; “****” indicates *p* < 0.0001.

**Figure 7 biomedicines-13-00792-f007:**
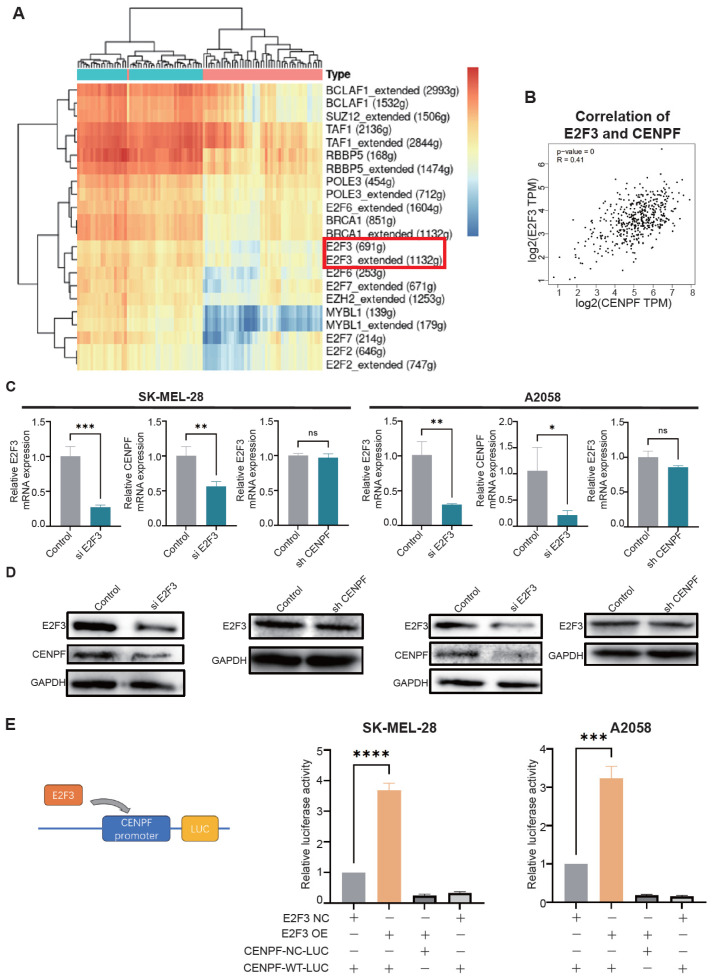
**Upregulation of CENPF was regulated by E2F3 in melanoma cells.** (**A**) SCENIC analysis prediction of the potential transcription factors of CENPF. (**B**) TCGA datasets prediction of a positive relationship between CENPF and E2F3. (**C**,**D**) qRT-PCR (**C**) and Western blot (**D**) assays showing the upregulation of E2F3 increased the expression of CENPF in two cell lines, but knockdown of CENPF did not influence E2F3 expression. (**E**) Luciferase reporter gene assay confirming that E2F3 could transcriptionally activate CENPF. “ns” indicates not significant; “*” indicates *p* < 0.05; “**” indicates *p* < 0.01; “***” indicates *p* < 0.001; “****” indicates *p* < 0.0001.

**Figure 8 biomedicines-13-00792-f008:**
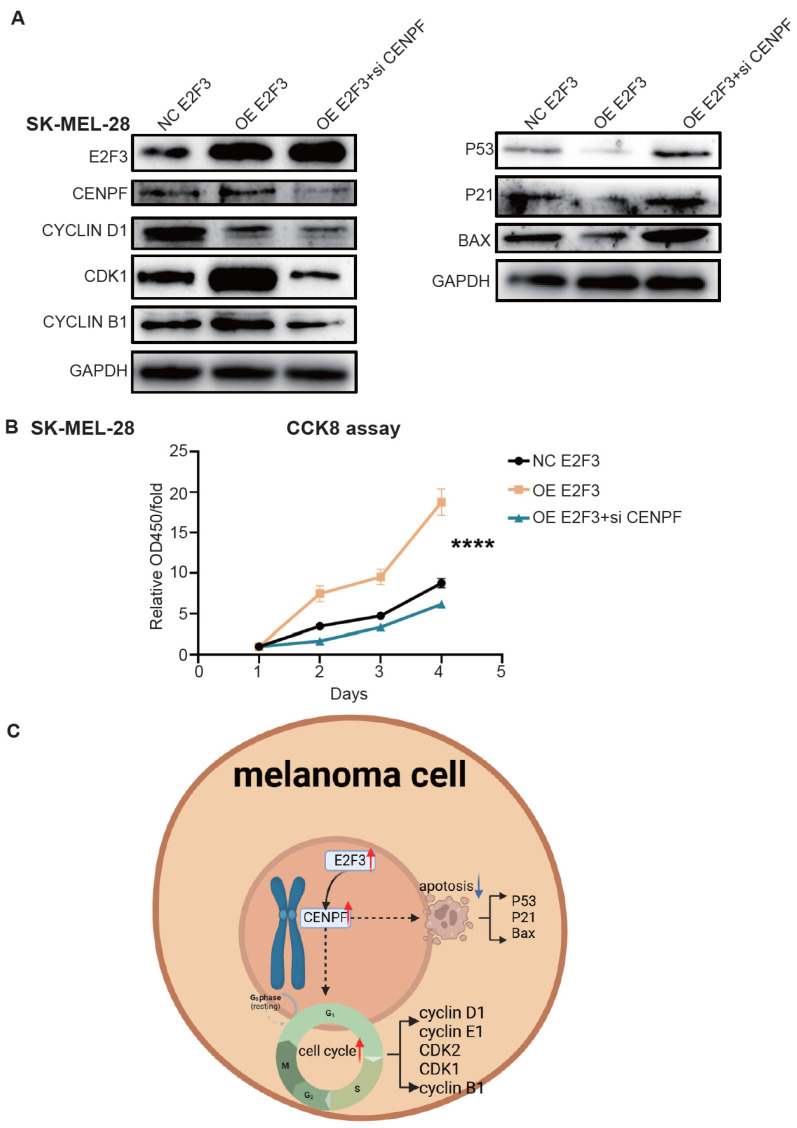
**Rescue experiments verified the effect of silencing CENPF in reversing the E2F3–CENPF axis.** (**A**) Western blot assays showing changes in G1 phase marker cyclin D1, G2/M markers CDK1 and cyclin B1, and P53 pathway proteins after over-expressing E2F3 or both over-expressing E2F3 and knocking down CENPF. (**B**) CCK-8 assay showing impaired proliferative ability after knocking down CENPF. (**C**) Model graph illustrating the biological function and mechanism of CENPF in melanoma. The red arrows represents upward process, while the blue arrow represent downward. “****” indicates *p* < 0.0001.

**Table 1 biomedicines-13-00792-t001:** **Correlations between CENPF with clinicopathologic features in 128 melanoma patients.** Note: A chi-square test was used for comparing groups between low and high CENPF expression. * *p* < 0.05 was considered significant.

Variable	Number of Patients		*p*-Value *
	CENPF-Low	CENPF-High	
Gender (n %)			0.1518923
female	39 (30.5%)	20 (15.6%)	
male	37 (28.9%)	32 (25%)	
Age, year			0.5275089
<60	32 (25%)	19 (14.8%)	
≥60	44 (34.4%)	33 (25.8%)	
Breslow depth (mm)			0.0424979
≤2	46 (35.9%)	22 (17.2%)	
>2	30 (23.4%)	30 (23.4%)	
Clark level			0.0062645
I–III	45 (35.2%)	18 (14.1%)	
IV–V	31 (24.2%)	34 (26.6%)	
Ulceration			0.0408965
Absent	16 (12.5%)	4 (3.1%)	
Present	60 (46.9%)	48 (37.5%)	
Lymph nodes metastasis			0.3157056
No	71 (55.5%)	45 (35.2%)	
Yes	5 (3.9%)	7 (5.5%)	
Distant metastasis			0.4412787
No	60 (46.9%)	38 (29.7%)	
Yes	16 (12.5%)	14 (10.9%)	
Clinical stage			0.2171654
I–II	56 (43.8%)	33 (25.8%)	
III–IV	20 (15.6%)	19 (14.8%)	

**Table 2 biomedicines-13-00792-t002:** **Univariate and multivariate analyses of** **factors associated with OS and DFS.**

	OS			
Variable	Univariate HR (95% CI)	Univariate P	Multivariate HR (95% CI)	Multivariate P
Gender (Women vs. Men)	1.276 (0.802–2.031)	0.304		
Age, year (<60 vs. ≥60)	1.236 (0.771–1.983)	0.379		
Breslow depth (mm) (≤2 vs. >2)	1.093 (0.689–1.732)	0.706		
Clark level (I–III vs. IV–V)	2.068 (1.290–3.316)	0.003	1.148 (0.653–2.021)	0.631
Ulceration (Absent vs. Present)	1.854 (0.917–3.746)	0.086		
Lymph nodes metastasis (No vs. Yes)	2.314 (1.184–4.524)	0.014	1.123 (0.308–4.090)	0.86
Distant metastasis (No vs. Yes)	2.644 (1.609–4.345)	<0.001	1.159 (0.352–3.821)	0.808
Clinical stage (I–II vs. III–IV)	3.249 (2.016–5.237)	<0.001	2.612 (0.740–9.222)	0.136
CENPF staining (low vs. high)	1.956 (1.222–3.132)	0.005	1.829 (1.114–3.002)	0.017
	**DFS**			
**Variable**	**Univariate HR (95% CI)**	**Univariate P**	**Multivariate HR (95% CI)**	**Multivariate P**
Gender (Women vs. Men)	1.245 (0.782–1.981)	0.356		
Age, year (<60 vs. ≥60)	1.209 (0.753–1.940)	0.432		
Breslow depth (mm) (≤2 vs. >2)	1.206 (0.761–1.912)	0.425		
Clark level (I–III vs. IV–V)	2.128 (1.324–3.421)	0.002	1.238 (0.704–2.178)	0.458
Ulceration (Absent vs. Present)	1.877 (0.931–3.784)	0.078		
Lymph nodes metastasis (No vs. Yes)	2.593 (1.322–5.088)	0.006	1.156 (0.322–4.157)	0.824
Distant metastasis (No vs. Yes)	2.442 (1.490–4.003)	<0.001	1.060 (0.320–3.517)	0.924
Clinical stage (I–II vs. III–IV)	3.190 (1.973–5.157)	<0.001	2.544 (0.712–9.095)	0.151
CENPF staining (low vs. high)	1.845 (1.155–2.946)	0.01	1.596 (0.973–2.620)	0.064

## Data Availability

The data presented in this study are openly available in National Center for Biotechnology Information, reference number GSE46517, GSE3189, GSE7553.

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
