# Peer review of "Centromere Protein F Is a Potential Prognostic Biomarker and Target for Cutaneous Melanoma"

_biomedicines, 2025, doi:10.3390/biomedicines13040792_

Round 1
Reviewer 1 Report
Comments and Suggestions for Authors
In the manuscript the authors investigate the role of CENPF in melanoma and suggests its role as a prognostic biomarker and therapeutic target in patients with this cancer. The manuscript is well written, most results confirm the main thesis of the research and proper conclusions are drawn from presented results. Materials and methods are generally well described, although I must say that the version I got for the review lacks tables and supplementary data, therefore I am unable to assess some of the results.
There are some issues that the authors should refer to.
Materials and methods:
- As far as I know SCENIC package is for single cell RNA-seq data (as the name of the package suggests). Is it possible to use bulk RNA-seq data from TCGA in this type of the analysis?
- There is an issue with the GSE45617 dataset number. This number directs to a research conducted on Staphylococcus aureus, not melanoma. Also, GSE3189 contains 18 nevi and only 7 normal samples, unlike stated in the manuscript (25 normal samples).
- Url addresses of all the websites or online tools used in the research should be given.
- The meaning of asterisks in statistical analysis should be explained in material and methods chapter, or in the legend of every figure in the manuscript.
Results and figures:
- The interpretation of FACS results is a bit misleading, e.g. when we see that in A2058 cell line the apoptosis is two more times effective than in control cells, it seems very convincing. Actually, when we analyze the real numbers, the apoptosis increases circa 1.5 % (in case of that particular cell line) which is rather marginal. The results in SK-MEL-28 are even worse. Therefore, when analyzing figure 5A we might suspect that silencing of CENPF mostly induces cell cycle arrest and subsequent cell death is not a result of massive apoptosis?
- lines 481-482: "Both overexpressing E2F3 and knockdown CENPF reversed the cell cycle-promoting effect of E2F3 and arrested cell in G2/M phase" - This sentence should be rephrased, because it suggested that E2F3 overexpression OR CENPF knockdown promote cell cycle arrest, individually. The overexpression of E2F3 itself obviously promotes cell cycle, which is confirmed by the results in Figure 8A and 8B. Only E2F3 overexpression followed by CENPF inhibition reverses this effect.
- Fig 1C – why the authors did not process the KEGG and GO enrichment analysis for 54 intersected genes from all three GEO datasets?
- Fig 1E lack of description of x axis (tumor/normal).
- Fig 2B is not a histogram but a bar plot. What is the unit of gene expression on y-axis here and in Fig 2A?
- SUZ12 is not a transcription factor.
- Figure 8A shows only WB of proteins involved in cell cycle control, and not the cell cycle checkpoints distribution itself. This is a big overstatement.
Author Response
Reviewer #1
Response: Thank you so much for your thorough and insightful feedback. Your feedback not only helped us correct the errors in the manuscript but also enhanced our thinking and approach regarding scientific research and future studies. Here below are our responses.
【Comment 1】Materials and methods are generally well described, although I must say that the version I got for the review lacks tables and supplementary data, therefore I am unable to assess some of the results.
Response: I am so sorry for the inconvenience caused. The supplementary tables and figures have been re-uploaded in the attachment titled “Tables and Supplementary Figures.” I hope they are now accessible for your review.
【Comment 2】As far as I know SCENIC package is for single cell RNA-seq data (as the name of the package suggests). Is it possible to use bulk RNA-seq data from TCGA in this type of the analysis?
Response: Thanks for your valuable question. Regarding the applicability of the SCENIC package to bulk RNA data, I understand that SCENIC infers transcriptional regulatory networks from RNA data. And in the 2020 paper by the SCENIC developers--Robust gene expression programs underlie recurrent cell states and phenotype switching in melanoma, both single-cell and bulk RNA data were employed to demonstrate SCENIC’s utility. In our work, SCENIC was applied to TCGA bulk RNA data, considering the overall RNA expression of a melanoma patient as one sample. While this approach may not be as precise as single-cell RNA data, it is still a feasible and informative method.
【Comment 3】There is an issue with the GSE45617 dataset number. This number directs to a research conducted on Staphylococcus aureus, not melanoma. Also, GSE3189 contains 18 nevi and only 7 normal samples, unlike stated in the manuscript (25 normal samples).
Response: I sincerely apologize for the error in the GEO dataset code. The correct code is GSE46517, and I have corrected this in the manuscript accordingly. And as noted in your comments, GSE3189 comprises 7 normal skin, 18 nevi, and 45 melanoma samples. In our analysis, both nevi and normal skin were treated as normal samples for comparison with melanoma samples. We explained this in the manuscript.
【Comment 4】Url addresses of all the websites or online tools used in the research should be given.
Response: Thank you for your I have updated the manuscript with the URLs of all the websites and online tools used.
【Comment 5】The meaning of asterisks in statistical analysis should be explained in material and methods chapter, or in the legend of every figure in the manuscript.
Response: The meaning of the asterisks in the statistical analysis has now been explicitly stated in the manuscript. In figures, we use different numbers of stars to represent different levels of significance: “*” indicates p < 0.05; “**” indicates p < 0.01; “***” indicates p < 0.001; “****” indicates p < 0.0001.
【Comment 6】The interpretation of FACS results is a bit misleading, e.g. when we see that in A2058 cell line the apoptosis is two more times effective than in control cells, it seems very convincing. Actually, when we analyze the real numbers, the apoptosis increases circa 1.5 % (in case of that particular cell line) which is rather marginal. The results in SK-MEL-28 are even worse. Therefore, when analyzing figure 5A we might suspect that silencing of CENPF mostly induces cell cycle arrest and subsequent cell death is not a result of massive apoptosis?
Response: As you correctly pointed out, CENPF knockdown resulted in a slight increase in apoptosis. The main effect of CENPF knockdown was the induction of cell cycle arrest. We presented this finding, which wanted to share with other researchers: CENPF knockdown alone induces cell cycle arrest but slight tumor cell death. This suggests that a combination of interventions targeting other pathways may be needed to achieve effective tumor cell eradication.
【Comment 7】lines 481-482: "Both overexpressing E2F3 and knockdown CENPF reversed the cell cycle-promoting effect of E2F3 and arrested cell in G2/M phase" - This sentence should be rephrased, because it suggested that E2F3 overexpression OR CENPF knockdown promote cell cycle arrest, individually. The overexpression of E2F3 itself obviously promotes cell cycle, which is confirmed by the results in Figure 8A and 8B. Only E2F3 overexpression followed by CENPF inhibition reverses this effect.
Response: Regarding the sentence on lines 481-482: "Overexpression of E2F3 and CENPF knockdown both reversed the cell cycle-promoting effect of E2F3 in G2/M phase-arrested cells," I recognize that the expression was unclear. What I wanted to convey is that after overexpression of E2F3, subsequent CENPF knockdown reverses the enhanced proliferative effect of E2F3 in melanoma, suggesting that CENPF is indeed a downstream molecule in the E2F3-mediated proliferative pathway.
【Comment 8】Fig 1C – why the authors did not process the KEGG and GO enrichment analysis for 54 intersected genes from all three GEO datasets?
Response: In Figure 1C, we performed GO and KEGG enrichment analyses on the 54 common genes (the GO and KEGG plots for the 54 genes are provided in the attachment). However, the results did not reveal significant pathways, likely due to the small number of genes included. Therefore, we performed enrichment analysis on the overlapping genes between two gene sets.
【Comment 9】Fig 1E lack of description of x axis (tumor/normal).
Response: In Figures 1E and Supplementary Figure 2, red boxes represent tumor samples, while black boxes represent normal samples. We added this explanation in the figure legends.
【Comment 10】Fig 2B is not a histogram but a bar plot. What is the unit of gene expression on y-axis here and in Fig 2A?
Response: Figure 2B has been corrected to indicate a bar chart, rather than a histogram. Regarding the gene expression data in Figures 2A and 2B, these were extracted from the original expression matrix. The original authors did not specify the units of measurement. Since our focus was on the differential expression of CENPF between melanoma patients and normal individuals, the specific units were not a primary concern. To avoid confusion, we have modified the y-axis labels to “relative expression levels”, thus preventing any misunderstanding regarding the units.
【Comment 11】SUZ12 is not a transcription factor.
Response: Thank so much for pointing this out. Concerning SUZ12 not being a transcription factor, it is indeed not. It appears in the SCENIC output because SCENIC results represent a set of genes and their associated regulatory factors, forming a "regulon." Therefore, the names listed in the output correspond to the regulons, which may not all be transcription factors. We have corrected this in the manuscript.
【Comment 12】Figure 8A shows only WB of proteins involved in cell cycle control, and not the cell cycle checkpoints distribution itself. This is a big overstatement.
Response: Thank so much for the very good question. Since determining the role of CENPF in melanoma is a central aspect of our study, we only employed Western blot (WB) to confirm the effects of E2F3 overexpression and CENPF knockdown on the cell cycle. However, as you suggested, flow cytometry would provide a more comprehensive validation of these findings. In future studies, we will design the experiment more carefully and validate our hypothesis from multiple perspectives.
Once again, I would like to express my sincere gratitude for your valuable comments. I hope the revisions meet your expectations, and I would be happy to address any further questions or suggestions you may have.

Reviewer 2 Report
Comments and Suggestions for Authors
This is a carefully performed study of the cell-cycle associated centromeric protein F (CENPF) in malignant melanoma cell lines and patient derived malignant melanoma and peritumoral tissue. The authors confirm and enhance that the identification of high CENPF expression is associated increased proliferation-index and shortened overall survival. On the other side in vitro and in vivo experiments show that downregulation of CENPF expression inhibits the proliferation and metastasis of melanoma cells. It appears that increased binding of the E2F3 transcription factor on the CENPF promoter is the main event responsible for high CENPF expression.
Overall, CENPF expression appears as a possible biomarker in malignant melanoma and opens the possibility that the E2F3-CENPF interaction represents a new therapeutic target in the treatment of this very aggressive cancer.
In summary well performed study and clear presentation.
Minor changes:
Line 51: (7-13), Line 83: 1 normal melanocyte cell line, Line 97: Lane 461: (30-33), Lane 462: in two cell lines.
Author Response
【Comment 1】Minor changes:
Line 51: (7-13), Line 83: 1 normal melanocyte cell line, Line 97: Lane 461: (30-33), Lane 462: in two cell lines.
Response: We are deeply grateful for your meticulous review and valuable suggestions. We have made the corresponding revisions in the manuscript one by one. Wishing you all the best in your work and life!